# Altered Expression of Interleukin-18 System mRNA at the Level of Endometrial Myometrial Interface in Women with Adenomyosis

Liang-Hsuan Chen [1,2], She-Hung Chan [3,4], Chin-Jung Li [1], Hsien-Ming Wu [1,2] and Hong-Yuan Huang [1,3,*]

1   Department of Obstetrics and Gynecology, Linkou Medical Center, Chang Gung Memorial Hospital, Taoyuan 33305, Taiwan
2   Graduate Institute of Clinical Medical Sciences, College of Medicine, Chang Gung University, Taoyuan 33302, Taiwan
3   Department of Obstetrics and Gynecology, College of Medicine, Chang Gung University, Taoyuan 33302, Taiwan
4   Department of Cosmetic Science, Providence University, No. 200, Sec. 7, Taiwan Boulevard, Shalu Dist., Taichung 43301, Taiwan
*   Correspondence: hongyuan@cgmh.org.tw

**Abstract:** Adenomyosis is a uterine pathology characterized by a deep invasion of endometrial glands and stroma, disrupting the endometrial–myometrial interface (EMI). Interleukin-18 (IL-18) system is a dominant cytokine involved in the menstrual cycle of human endometrium. IL-18 may play a defensive role against maternal immune response in the uterine cavity. The present study was designed to determine IL-18-mediated immune response at the level of EMI. We uncovered that mRNA of IL-18 system, including IL-18, IL-18 receptor (IL-18R), and its antagonist, IL-18 binding protein (IL-18BP), expressed in eutopic, ectopic endometrium, and corresponding myometrium in patients with adenomyosis. IL-18 system was demonstrated in paired tissue samples by immunochemistry and immunofluorescence study. According to RT-PCR with $C_T$ value quantification and $2^{-\Delta\Delta Ct}$ method, a significant down-regulation of IL-18BP in corresponding myometrium in comparison to eutopic endometrium ($p < 0.05$) indicates that the IL-18 system acts as a local immune modulator at the level of EMI and regulating cytokine networks in the pathogenesis of adenomyosis. Furthermore, an increased IL-18 antagonist to agonist ratio was noted in ectopic endometrium compared with corresponding myometrium. We suggest that altered IL-18 system expression contributes to immunological dysfunction and junctional zone disturbance in women with adenomyosis.

**Keywords:** interleukin-18; cytokine; adenomyosis; endometrium; PCR

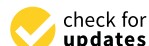



## 1. Introduction

Endometriosis, a gynecologic disorder, occurs in about 10–15% of women of reproductive age and is associated with subfertility [1]. Endometriosis is characterized by the endometrial glands and stroma that normally line inside the uterus, growing outside the uterus, commonly including ovaries and peritoneum [2]. Among the etiologies of endometriosis, the current consensus is that retrograde menstruation has been implicated in the peritoneal seeding of endometrial tissue [3]. This dissemination of ectopic endometrial implants activates humoral- and cell-mediated inflammation. An increased number of activated macrophages and lymphocytes are found in the peritoneal cavity of women with endometriosis, which plays a crucial role in survival and invasion of the ectopic endometrium [4].

Adenomyosis is a related uterine disease presented with ectopic endometrium surrounded by hypertrophic and hyperplastic myometrium, producing a diffuse enlargement of the uterus [5,6]. Patients with adenomyosis often suffer from severe dysmenorrhea

and/or menorrhagia. Although adenomyosis is thought to be closely related to endometriosis, adenomyosis and endometriosis are identified by the presence of endometriotic tissues inside or outside the uterus, respectively [7]. Compared with endometriosis regarding clinical features and pathogenetic mechanisms, adenomyosis is more common in multiparous women and may cause a higher possibility of abortion [8,9].

The pathogenesis of adenomyosis has been explained in several theories [10]. The most widely accepted mechanism of adenomyosis is a deep invasion of the inner myometrium by eutopic endometrium disrupting the endometrial–myometrial interface (EMI) [11,12]. Consequently, a series of molecular and metabolic disturbances shows in eutopic and ectopic endometria of women with adenomyosis, which stimulates angiogenesis and proliferation, impairs apoptosis, produces estrogens, increases progesterone resistance, and downregulates cytokine expression. The endometrial tissue infiltrates the junctional zone myometrium, and the ectopic endometrial glands grow afterward [10,13].

Previous studies have demonstrated that disturbances in the immune system give rise to endometriosis [14–16]; however, no similar reports have indicated the immune modulation of adenomyosis, particularly regarding immunosuppression. Accumulating evidence declares human and cellular immunity alterations in adenomyosis [7,17]. In addition, many different types of immune-reacted serum proteins were proposed to develop the pathologic process of adenomyosis [18]. Recent expeditious understanding of the immunomodulation of interleukin-18 (IL-18) throughout the reproductive system has been reported in the menstrual cycle [19,20].

IL-18, a member of the interleukin-1 (IL-1) family [21–23], was described in 1995 [24] as a cofactor for both Th1 and Th2 cells [25]. IL-18, like IL-1, is an 18-kDa proinflammatory cytokine generated from pro-IL-18 (a 24-kDa precursor) by the cysteine protease caspase-1 [26,27]. The structure of IL-18 is closely associated with IL-33 and IL-1β; however, the function of IL-18 is quite different from that of IL-1. IL-18 was initially discovered in endotoxin-challenged mice and identified as a circulating molecule stimulating macrophages and interferon-γ inducing factor (INF-γ) [24]. In collaboration with IL-12, IL-18 activates the Th1-mediated immune response, participating in the host defense against infection. IL-18 alone can trigger the Th2 response resulting in an allergic reaction. Activated Th2 cells produce IL-4, IL-5, IL-10, and IL-13, leading to immunoglobulin production by B cells and eosinophil activation [28,29]. To recap, IL-18 can activate innate immunity and both Th1- and Th2-mediated responses.

IL-18 system is a group of polypeptides comprised of IL-18, IL-18 receptor (IL-18R), and IL-18 binding protein (IL-18BP). The binding sites for IL-18 to IL-18R are identical to those for IL-1 binding to IL-1 receptor type I [30]. IL-18BP was initially found in human urine and served as a natural inhibitor of IL-18-induced IFN-γ, suppressing Th1-mediated response [31]. The alterations of IL-18 system may participate in the pathologic process of endometriosis [32–35]. IL-18 may regulate proinflammatory cytokines, particularly IL-8 and TNF-α, and act as a potent angiogenic factor by inducing endothelial cell migration and increasing matrix metalloproteinases production [32]. Functional IL-18, generated in situ, converts the microenvironment into a Th2 dominant status in endometriotic cells, leading to local autoimmunity [33]. The elevation of peritoneal IL-18 was found in endometriosis patients with macrophage infiltration and COX-II induction, leading to prostaglandins production, and eventually causing pain. In contrast, IL-18 has been reported to affect endometriosis negatively [34,35]. Suppression of IL-18 in women with endometriosis contributes to decreased natural killer (NK) cell activity which enhances the opportunity for escaping immune elimination in ectopic endometrium, resulting in survival and growth of ectopic endometrial implants [35].

To date, there has been little information regarding the nature of IL-18 system in the pathophysiologic mechanism of adenomyosis. We have previously demonstrated that the expression of eutopic endometrial IL-18 system and the ratio of IL-18 antagonist to agonist at the level of EMI. The previous study showed a higher ratio of IL-18 antagonist to agonist in patients with adenomyosis than in healthy controls, indicating that dysregulation of

IL-18 system possibly contributed to the pathogenesis of adenomyosis [36]. However, IL-18-mediated immune response remains obscure in the process of adenomyosis. The present study was designed to determine whether IL-18 expresses in the ectopic and/or eutopic endometrium, and to compare IL-18 system expression among the eutopic endometrium, ectopic endometrium, and corresponding myometrium in patients with adenomyosis.

## 2. Materials and Methods

### 2.1. Ethics Statement and Tissue Collection Protocol

Paired tissue samples with their homologous eutopic endometrium, ectopic endometrium, and corresponding normal myometrium were collected from 10 premenopausal women undergoing hysterectomy. The study was approved by the Chang Gung Memorial Hospital institutional review board (CGMG IRB#200701502B0), and written informed consent was obtained from all participants. The indications for hysterectomy included clinical symptoms associated with adenomyosis, such as hypermenorrhea and dysmenorrhea. None of the participants had been diagnosed with pelvic inflammatory disease or genital tract infection, and none had received any hormone therapies within six months before surgery. All biopsies were performed by an experienced gynecologist at Chang Gung Memorial Hospital, an academic medical center in Taoyuan, Taiwan. The excised tissue samples were histologically documented. The presence of adenomyosis foci was confirmed to include active endometrial glands and stroma surrounded by hypertrophic myometrium. We used the human placenta as a positive control for IL-18 system in the PCR and immunohistochemistry studies.

### 2.2. Specimen Treatments

Fresh tissue specimens were divided into two parts. One part was fixed in 4% formaldehyde, embedded in optimal cutting temperature compound (OCT; Shakura Finetek, Torrance Inc., Torrance, CA, USA), and frozen in liquid nitrogen until sectioned. The other part was immediately stored at $-80\ ^\circ$C until RNA extraction. To study mRNA expression of IL-18 system, total RNA was extracted with RNAzol reagent (Tel-test, Inc., Friendswood, TX, USA) as previously described [37]. RNA concentration was quantified by measuring optical density with a Spectronic 601 spectrophotometer (Milton Roy Co., Rochester, New York, NY, USA). For immunohistochemistry staining, frozen eutopic endometrium with attached myometrium tissue was excised in twelve serial sections (6 μm) from each sample. The first and the last slides were stained with hematoxylin-eosin for pathologic conformation, and the subsequent slides stained for IL-18 system were examined by a Nikon microphot-FXA microscope (Nikon Instruments, Garden City, New York, NY, USA).

### 2.3. Reverse Transcription (RT) and PCR

Complementary DNA was synthesized from 1 μg of total RNA. The reaction was carried out in a total volume of 20 μL of RT-Master Mix in a 0.5 mL thin wall PCR tube (Applied Scientific, South San Francisco, CA, USA). Specific sequences of oligonucleotide primers for β-actin (838 bp), IL-18 (385 bp), IL-18 R (157 bp), and IL-18BP (240 bp) target cDNA were used for PCR amplification as described previously [20]. We used the GeneAmp RNA PCR kit (Perkin-Elmer, Foster City, CA, USA), and the reverse transcription (RT) was performed in the DNA Thermal Cycler 480 (Perkin-Elmer GeneAmp, PCR Instrument System, Branchburg, NJ, USA). The program consisted of the following steps: 15 min at $42\ ^\circ$C, 5 min at $99\ ^\circ$C, and then quenched at $4\ ^\circ$C. RT products were stored at $-20\ ^\circ$C before the subsequent PCR. The corresponding paired primers were added to the RT products in the PCR Master Mix to a total volume of 100 μL. PCR was performed simultaneously from a single Master Mix in different tubes with each primer. PCR cycles were composed of one process of 5 min at $95\ ^\circ$C to denature all proteins, 30 cycles of 60 s at $94\ ^\circ$C, 60 s at $60\ ^\circ$C, and 60 s at $72\ ^\circ$C. The reaction was terminated at $72\ ^\circ$C for 5 min and quenched at $4\ ^\circ$C.

### 2.4. Agarose Gel Electrophoresis

The 2% agarose gel (Gibco BRL) was stained with ethidium bromide (Sigma Chemical Co., St. Louis, MO, USA) and performed in an H5 electrophoresis chamber. Each PCR product (25 μL) was carried out in parallel with a standard DNA ladder (Gibco BRL). After electrophoresis completion, the gel blot and printed photocopies of the blot were calculated on the UV-densitometry (Alpha ImagerTM IS-2200 system, Alpha Innotech, San Diego, CA, USA).

### 2.5. mRNA Expression Determined by Quantitative PCR (Q PCR)

The gene expression of IL-18, IL-18R, and IL-18BP in paired uterine samples was measured by real-time quantitative PCR, based on the TaqMan methodology using Applied Biosystems 7900 HT Real-time PCR System. According to the manufacturer's instructions, PCR reactions were processed to a final volume of 25 μL, containing 12.5 μL of 2X TaqMan Universal PCR Master Mix (Life Technologies®, Foster City, CA, USA), 1.25 μL TaqMan assay (20×), 1μL of sample cDNA, and 10.25 μL of RNAse-free water.

### 2.6. $C_T$ Value Quantification and $2^{-\Delta\Delta Ct}$ Method

The amount of IL-18, IL-18R, and IL-18BP mRNA in eutopic and ectopic endometrium were compared with the respective amounts of myometrium in the same patient. The relative transcript level was analyzed by the $2^{-\Delta\Delta Ct}$ method taking GAPDH as the standard internal control [38,39].

### 2.7. Immunohistochemistry and Immunofluorescence of IL-18 System in Adenomyosis

To identify IL-18 system proteins at EMI in patients with adenomyosis, samples were prepared for immunohistochemical study as described previously [20]. Human placental tissue was used as a positive control. Omitting the primary antibody served as a negative control. EMI tissues were cut at 6 μm of serial frozen sections, fixed in 4% paraformaldehyde for 15 min, washed with 0.05% Tween 20/Phosphate-buffer saline for 10 min with shaking, and then incubated with 3% $H_2O_2$ (DAKO, Glostrup, Denmark) in PBS for 10 min to eliminate endogenous peroxidase activity. Primary antibodies, including human monoclonal IL-18 antibody (R&D System, Minneapolis, UK), human IL-18R polyclonal antibody (R&D System), and human IL-18BP monoclonal antibody (Novus Biologicals, Littleton, CO, USA), were applied and diluted 1:50 in phosphate-buffer saline-BSA at 4 °C overnight. The negative control was incubated in phosphate-buffered saline-BSA only. The EnVision kit carried out immunohistochemical detection for IL-18 system with horseradish peroxidase (HRP) (Dako, Glostrup, Denmark) system as the link, and liquid 3, 3-Diaminobenzidine (DAB) as chromagen. Brown granules exhibited a positive antibody reaction. All sections were counterstained with Mayer hematoxylin, dehydrated, and mounted.

Immunofluorescence studies at an endogenous level were also carried out in paired tissues by staining for IL-18 system. After PBS rinses, sections were incubated with primary antibody diluted 1:50 in phosphate-buffer saline-BSA at 4 °C overnight, followed by a secondary antibody incubation [FITC-conjugated anti-rabbit IgG (Millipore Bioscience Research Reagents)] for IL-18 system detection and incubated in 1 mg/mL DAPI for nuclear staining. The tissues were examined by an Olympus BX53 fluorescence microscope.

### 2.8. Statistical Analysis

All values are expressed as the mean ± standard error. The statistical analysis was calculated by the SPSS 22.0 (SPSS Inc., Chicago, IL, USA), with a *p*-value of <0.05 considered statistically significant.

## 3. Results

### 3.1. Expression of IL-18 System at the Level of Endometrial–Myometrial Interface of Patients with Adenomyosis

We demonstrated IL-18, IL-18R, and IL-18BP mRNA expression in eutopic, ectopic endometrium, and corresponding myometrium. Real-time quantitative PCR in paired

samples was performed. According to $C_T$ value quantification and the $2^{-\Delta\Delta Ct}$ method, the amounts of IL-18, IL-18R, and IL-18BP mRNA in eutopic and ectopic endometrium were compared with the respective amounts of myometrium in the same patients (Table 1). The expression of IL-18 was not significantly different in eutopic, ectopic endometrium, and corresponding myometrium. IL-18R was significantly lower in ectopic endometrium and corresponding myometrium than in eutopic endometrium ($p < 0.05$). As an antagonist of IL-18, IL-18BP was significantly lower in corresponding myometrium than in eutopic endometrium ($p < 0.05$). Furthermore, an increased ratio of IL-18 antagonist to agonist in ectopic endometrium compared with corresponding myometrium demonstrated that IL-18 system may play an essential role in regulating immune response and suppressing the rejection of eutopic endometrium implantation at the level of EMI (Figure 1).

**Table 1.** Expression of IL-18 system mRNA.

| Paired Tissue ($n = 10$) | mRNA (Delta Ct) | | |
|---|---|---|---|
| | IL-18 | IL-18BP | IL-18R |
| Endometrium | | | |
| Eutopic | 8.2 ± 0.9 (1.0) | 11.8 ± 1.2 (1.0) | 7.6 ± 2.4 (1.0) |
| Ectopic | 8.0 ± 1.3 (1.1) | 12.0 ± 1.3 (0.2) | 9.8 ± 1.8 [b] (0.9) |
| Myometrium | 7.7 ± 1.3 (1.4) | 14.0 ± 2.9 [a] (0.3) | 9.5 ± 1.5 [c] (0.2) |

Results are expressed as mean ± SD (normalized data relative to eutopic endometrium). [a,b,c] $p < 0.05$.

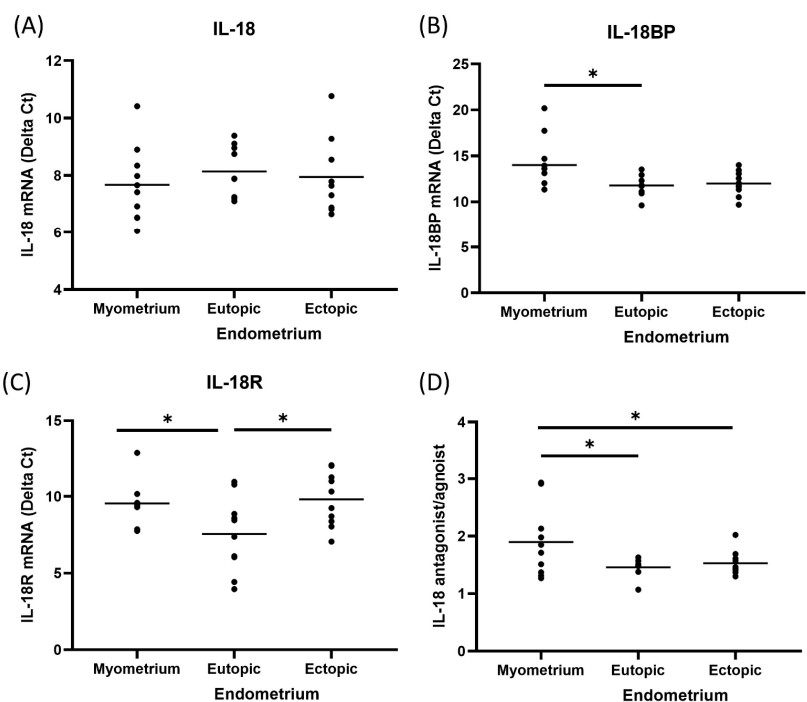

**Figure 1.** IL-18 mRNA determined by real-time quantitative PCR in paired tissue of patients with adenomyosis. According to $C_T$ value quantification and $2^{-\Delta\Delta Ct}$ method, (**A**) IL-18 was similar among the tissues from eutopic, ectopic endometrium, and corresponding myometrium; (**B**) IL-18BP was significantly lower in corresponding myometrium than those in eutopic endometriu; (**C**) IL-18R was significantly lower in ectopic endometrium and corresponding myometrium in comparison to eutopic endometrium; (**D**) an increased ratio of IL-18 antagonist to agonist in ectopic endometrium compared with corresponding myometrium is demonstrated that IL-18 system may play a critical role in regulating immune response and suppressing the rejection of eutopic endometrium implantation at the level of EMI. * $p < 0.05$.

### 3.2. Immunohistochemistry of IL-18 System in Patients with Adenomyosis

To demonstrate the presence of IL-18 system proteins at the level of EMI, formalin-fixed tissues were processed for immunohistochemical staining. Immunoreactive proteins of IL-18, IL-18R, and IL-18BP were identified among the eutopic endometrium, adenomyotic foci, and corresponding myometrium. Using horseradish peroxidase staining, the human placenta was used as a positive control, whereas negative control was presented by matched tissues omitting the primary antibodies (Figure 2). IL-18 expression was detected in both the glands and stroma of eutopic (Figure 2A) and ectopic (Figure 2B) endometrium. The epithelial cells showed higher staining intensity than the stromal cells in both eutopic and ectopic samples.

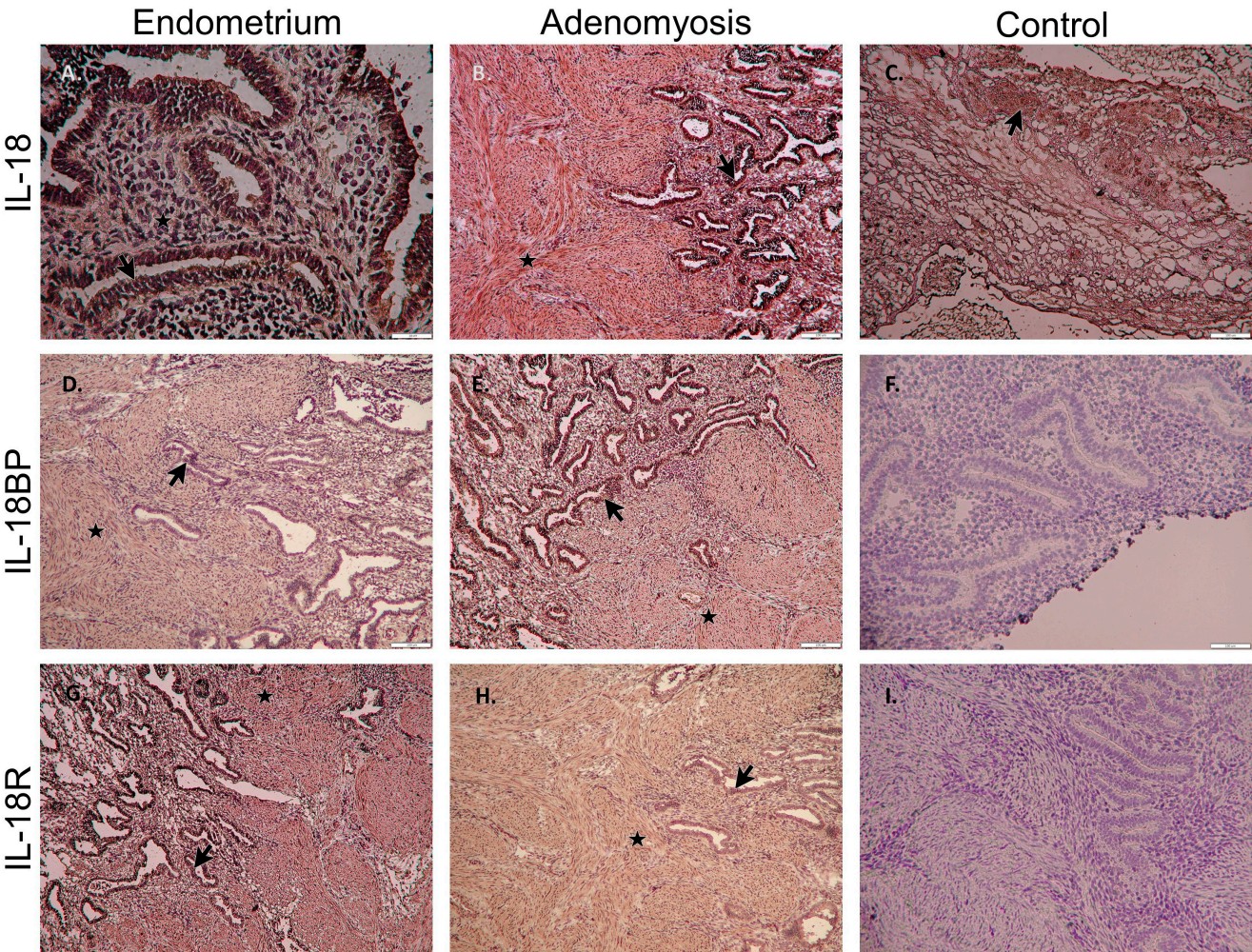

**Figure 2.** Immunohistochemistry of IL-18 system in patients with adenomyosis. A brown stain indicated a diffuse IL-18 system positive reaction in luminal epithelium (arrow) and stromal cells (star) of eutopic (**A**,**D**,**G**) and ectopic (**B**,**E**,**H**) endometrium. The human placenta was a positive control of IL-18 (**C**). Negative control was also presented by omitting the primary antibodies of IL-18BP (**F**) and IL-18R (**I**). (**A**) scale bar = 20 μm; (**B–I**) scale bar = 100 μm.

### 3.3. Immunofluorescence of IL-18 System in Patients with Adenomyosis

To better understand the potential involvement of IL-18 in the pathogenesis of adenomyosis, immunofluorescence studies at the endogenous level were also carried out in paired tissues by staining for IL-18 system (Figure 3).

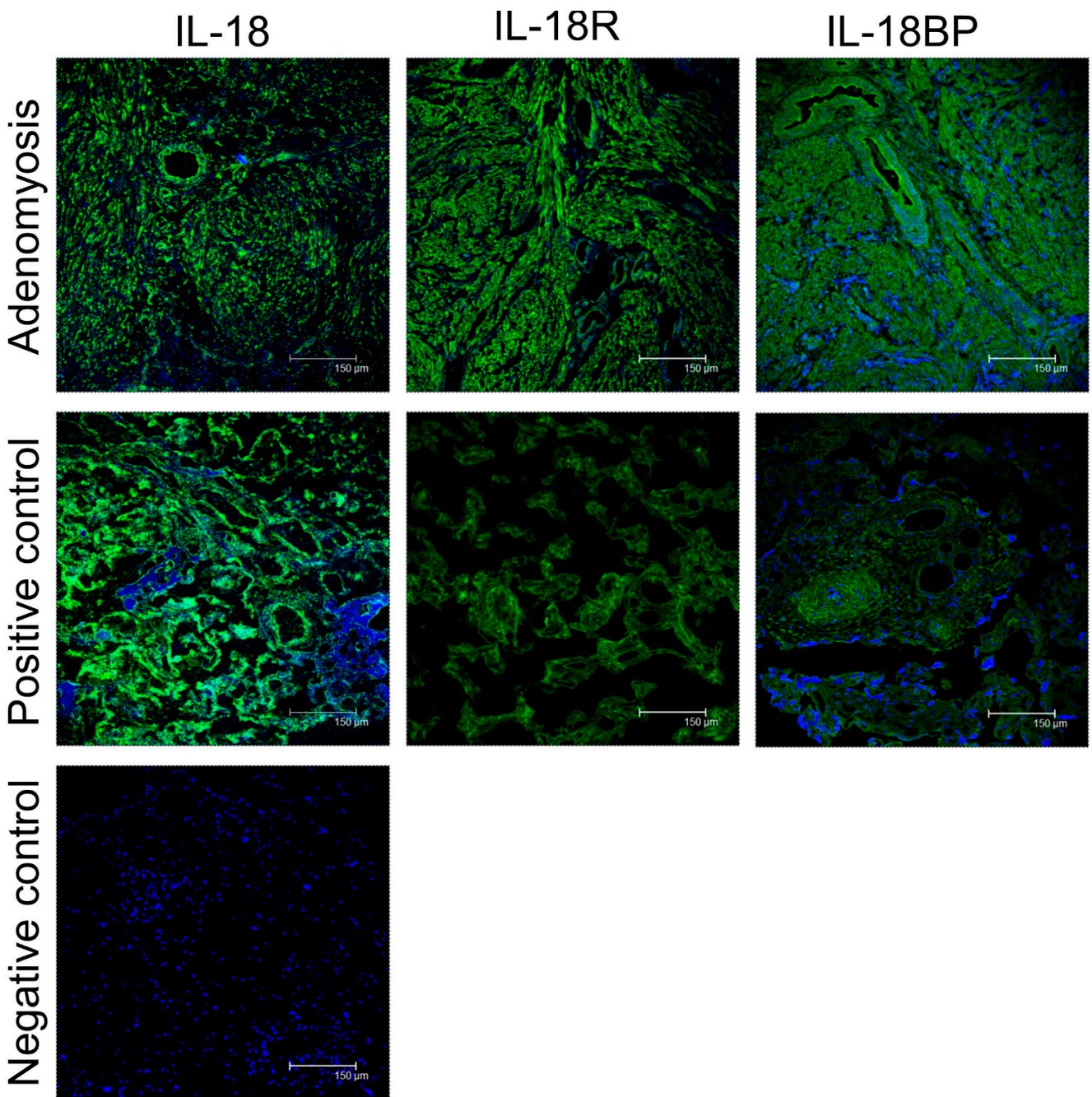

**Figure 3.** Immunofluorescence of IL-18 system in patients with adenomyosis. The expression of IL-18 system (green) is located in the cytoplasm of both glandular and stromal cells. The human placenta was a positive control of IL-18. The study samples from patients with adenomyosis showed significant overexpression of IL-18 system proteins, indicating the presence of IL-18 system may act as a local immunomodulator in adenomyosis. Scale bars = 150 μm.

## 4. Discussion

In human endometrium, a complex interaction of a network of cells is exhibited as an orchestrated phase of proliferation, differentiation, and menstrual shedding [40,41]. Cytokine production, expression of the receptors for cytokines, and regulation of endometrial functions by these factors reveal the critical role of cytokines in serving at autocrine, paracrine, and endocrine levels in the endometrium. Emerging evidence strongly suggests the involvement of proinflammatory cytokines in physiologic and pathologic processes, such as trophoblast invasion and endometriosis [42,43]. Our previous study declared that a

proper ratio of IL-1 agonist to receptor antagonist at embryo–epithelial interface might be crucial to initiating and preserving successful implantation [20,44]. In the present study, IL-18 and IL-18BP mRNA were detected in eutopic, ectopic endometrium, and corresponding myometrium at the level of EMI. With $C_T$ value quantification and the $2^{-\Delta\Delta Ct}$ method, significant down-regulation of IL-18BP in myometrium compared to eutopic endometrium ($p < 0.05$) implicates that IL-18 system acts as a local immune regulator at the level of EMI in adenomyosis.

In patients with endometriosis, the endometriotic implants and other immune cells in the peritoneum may produce high concentrations of cytokines, growth factors, and angiogenic factors in peritoneal fluid [43,45]. These chemotactic factors recruit macrophages and T lymphocytes, which mediate a local, sterile inflammatory response associated with the development of ectopic endometrium. Among cytokines derived from macrophages, the IL-1 family, especially IL-1β and IL-18, are strongly considered an immune modulator of the inflammatory process in the peritoneal cavity [46]. IL-1 and IL-18 are structurally homologous proteins and, together with their receptors, are members of the IL-1R/toll-like receptor (TLR) superfamily, with similar signaling pathways [21]. As a potent IFN-γ inducing factor, IL-18 participates in the host defense and tissue injury [25,47]. Previous references have shown that IL-18 regulates an immunomodulatory process in macrophages, T cells, B cells, keratinocytes, dendritic cells, astrocytes, and endometrial cells [19,25,47]. Particularly in the presence of IL-12, IL-18 induces the Th1 response and stimulates the production of cytokines (IFN-γ, IL-2, IL-15, GM-CSF, and TNF-α) from T cells and NK cells [24,25,47]. Indeed, IL-18R is expressed selectively on Th1 cells and can stimulate the production of IFN-γ [48]. In vivo, IL-18 is not only essential to host defense against severe infection but also crucial to tumor rejection by augmenting the cytotoxic activity of T and NK cells [25]. Besides, IL-18 alone can initiate a Th2 response from basophils and mast cells to induce an allergic reaction [25,47]. As described above, IL-18 is a potent cytokine that takes part in both Th1-and Th2-driven immune responses, resulting in multiple physiologic and pathologic processes. IL-18 was identified in human endometrium, indicating that IL-18 may provide protection against pathogenic microorganisms and modulate a local cytokine network for trophoblast invasion [19]. We have demonstrated the mRNA expression and protein production of a complete IL-18 system in both proliferative and secretory phases of human endometrium [20]. A significantly higher ratio of IL-18BP to IL-18 was documented in the secretory phase than in the proliferative phase, which may be responsible for embryo implantation in secretory endometrium.

Several cytokines and growth factors have been postulated to induce either internal or external types of endometriosis [49]. Although the endometrial cytokine profiling in patients with endometriosis has been illustrated, the identity of the receptors and their target cells for cytokine actions remains to be defined [50,51]. Some studies demonstrated a significantly lower concentration of IL-18 in the peritoneal cavity of women with endometriosis, which may indicate a deterioration of the Th1 response in the pathogenesis of endometriosis [35,46]. On the contrary, overexpression of IL-18 in peritoneal fluid and IL-18-induced COX-II in peritoneal monocytes [33] are reported in endometriosis patients; moreover, IL-18 level is higher in patients with minimal- to mild-stage peritoneal endometriosis [32]. Although some studies did not find significant differences in serum or peritoneal IL-18 levels between patients and healthy controls [52,53], a positive correlation between serum and peritoneal IL-18 concentrations in patients with endometriosis possibly suggests a systemic immunomodulatory role of IL-18 rather than solely peritoneal involvement of IL-18 [53].

Adenomyosis, an internal type of endometriosis, is a disorder in which the endometrial cells disrupt the EMI and scatter haphazardly within the myometrium [5,54]. The initiation of ectopic endometrium implantation with subsequent myometrial hypertrophy and hyperplasia around the adenomyosis foci at the EMI is still unclear. Adenomyosis is regarded as a junctional zonal disease, predisposed to the secondary disruption of the basal endometrial layer resulting in subendometrial smooth muscle hyperplasia and loss of

inner myometrial function [55]. From the immunologic basis, alteration in both cellular and humoral immunity contributes to adenomyosis, including the recruitment of immunomodulatory cells and deposition of inflammatory substances [7]. Elucidation of IL-18 system modulating at the level of EMI may provide new insights into the pathogenesis of adenomyosis. In a previous study, down-regulation of ectopic and eutopic endometrial IL-18 was found in patients with endometriosis, indicating that IL-18 might participate in the process of endometriosis [35]. The expression of endometrial IL-18R and the ratio of IL-18 antagonist to agonist are significantly higher in patients with adenomyosis than the healthy control [36]. Taken together with the present results, dysregulation of IL-18-mediated immune response (Figure 1) may interfere with the integrity of the junctional zone resulting in ectopic endometriotic lesions formation.

Ectopic endometrial implants were thought to be eradicated by immunosurveillance. The concept that defective immune response results in endometriotic lesion escape became more widespread. However, it remains obscure what affects the immune imbalance in adenomyosis. Several molecular mechanisms have been pointed out for endometriosis, including inhibiting TNF-$\alpha$-induced cellular apoptosis and enhancing NALP-3-induced IL-1$\beta$ production [56]. The overexpression of MMPs induced by IL-1$\beta$ via NF-kB activation provides endometriotic cell matric adhesion and invasion [57]. It is known that estrogen receptor $\beta$ and the NALP-3 inflammasome remarkably rise in endometriotic tissue, block the TNF-$\alpha$-induced cell death program, and trigger the activation of IL-1$\beta$ and IL-18 which promotes further inflammatory process, resulting in the adhesion and proliferation of ectopic endometrial cells [58]. The latest study also demonstrated the role of NLRP3/caspase-1/IL-1$\beta$-mediated pyroptotic pathway in endometriosis [59]. Some potential targets for novel anti-inflammatory therapies have been reported. Alpha lipoic acid, an antioxidant agent, has recently been highlighted to reduce the proinflammatory cytokine levels (e.g., IL-1$\beta$, IL-18) and adhesion molecule (e.g., ICAM-1) expression by inhibiting NALP-3 activity [58].

## 5. Conclusions

In summary, this study has illustrated the expression of the complete IL-18 system and its regulation at the level of EMI in eutopic, ectopic endometrium, and corresponding myometrium of patients with adenomyosis. We further demonstrate a lower expression of IL-18BP may interfere with the integrity of the junctional zone resulting in the invasion of the endometrium. Disruption of the EMI either directly by endometrial factors or indirectly by an IL-18-mediated immune response may contribute to a local immune-modulating cytokine network in the pathogenesis of adenomyosis.

**Author Contributions:** Conceptualization, H.-Y.H. and H.-M.W.; formal analysis, S.-H.C.; data curation, C.-J.L.; writing—original draft preparation, L.-H.C. and H.-Y.H.; writing—review and editing, H.-Y.H. All authors have read and agreed to the published version of the manuscript.

**Funding:** This study was supported in part by grants from [Chang Gung Memorial Hospital, Taiwan], Research Grant No. [CMRP37133, CMRPG3E0771, and CMRPG3K0781/3K0782].

**Institutional Review Board Statement:** The study was approved by the Institutional Review Board of Chang Gung Memorial Hospital (CGMG IRB#200701502B0).

**Informed Consent Statement:** Written informed consent was obtained from all subjects involved in the study.

**Data Availability Statement:** The data presented in this study are available on request from the corresponding author.

**Conflicts of Interest:** The authors declare no conflict of interest.

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
