# Peer review of "Altered Expression of Interleukin-18 System mRNA at the Level of Endometrial Myometrial Interface in Women with Adenomyosis"

_cimb, doi:10.3390/cimb44110376_

Round 1

Reviewer 1 Report

In this manuscript entitled “Altered expression of interleukin-18 system mRNA at the level of endometrial myometrial interface in women with adenomyosis”, the authors assess IL-18 system expression in patients with adenomyosis.

There are concerns about the quality of the data presented in the manuscript. In particular, the mRNA results do not seem to be supported by the immuno-antibody staining experiments.

Comments:

1. In line 92, the author's previous papers should be presented in detail. They should write what was elucidated and what problems were highlighted as a result. In other words, the purpose of the experiment in this paper is not clear.

2. The authors should clarify what kind of fluorescence microscope used.

3. In line 219, the authors should specify for what purpose they performed the experiment in Figure 2.

4. In Figure 2, please illustrate with arrows or other means the areas and positions where antibody reactions are observed.

5. In Figure 2, are the same magnifications used in the panels? A scale bar is needed.

6. What morphological features are seen in endometrium and adenomyosis? Also, in which areas is IL-18 typically expressed? Please clarify.

7. The data presented in the Figure 2 need to be quantified.

8. The legend in Figure 3 should be presented in detail.

9. All immunofluorescence data are of poor quality. Some panels are not in focus.

10. The authors need to clarify what they are trying to assert from the results in Figure 3.

11. In Discussion, the experimental results obtained in this study should be discussed. Please add.

Author Response

Point-by-point response to Reviewer 1

Thank you for the valuable comments. All of the changes we made in the manuscript were marked in red.

Comments:

  1. In line 92, the author's previous papers should be presented in detail. They should write what was elucidated and what problems were highlighted as a result. In other words, the purpose of the experiment in this paper is not

Ans: Thank you. We have highlighted the results of the previous study to clarify the present study’s purpose.

  1. The authors should clarify what kind of fluorescence microscope used. Ans: Thank you. The tissues were examined by an Olympus BX53 fluorescence
  2. In line 219, the authors should specify for what purpose they performed the experiment in Figure 2.

Ans: Thank you. In figure 2, we demonstrated the presence of IL-18 system proteins at the level of EMI. The purpose was addressed in the revised manuscript.

  1. In Figure 2, please illustrate with arrows or other means the areas and positions where antibody reactions are

Ans: Thank you. The IL-18-positive reaction was illustrated with arrows in the revised manuscript.

  1. In Figure 2, are the same magnifications used in the panels? A scale bar is

Ans: Thank you. A scale bar was added in the revised manuscript.

  1. What morphological features are seen in endometrium and adenomyosis? Also, in which areas is IL-18 typically expressed? Please

Ans: Thank you. Ectopic endometrium is defined as the presence of islands of endometrial glands and stroma within the myometrium. A representational morphology of eutopic and ectopic endometrium is

shown in Figure 2(A) and 2(B), respectively. IL-18 system expression was significantly localized in the cytoplasm of glandular and stromal cells. The statement was added to the revised manuscript.

  1. The data presented in Figure 2 need to be

Ans: Thank you. The epithelial and stromal staining was mainly cytoplasmic. The staining intensity score with a value of 2-3 was found. The endometrium epithelial cells showed higher staining intensity than the stromal cells in both eutopic and ectopic samples. The statement was added to the revised manuscript.

  1. The legend in Figure 3 should be presented in

Ans: Thank you. The legend in detail was provided in the revised manuscript.

“The expression of IL-18 system (green) is located in the cytoplasm of both glandular and stromal cells. The study samples from patients with adenomyosis showed significant overexpression of IL-18 system proteins, indicating the presence of IL-18 system may act as a local immunomodulator in adenomyosis.”

  1. All immunofluorescence data are of poor quality. Some panels are not in focus.

Ans: Thank you. We have revised the pictures.

  1. The authors need to clarify what they are trying to assert from the results in Figure 3.

Ans: Thank you. It will be provided in the legend of Figure 3 in detail. The immunofluorescence data confirmed the presence of IL-18 system in study samples. The RT-PCR method provided the qualification test of IL-18 system (Figure 1 and Table 1).

  1. In Discussion, the experimental results obtained in this study should be Please add.

Ans: Thank you. The results presented in this study was addressed in Line 259-264, and further discussed in Line 314-322.

Thank you again for all the thoughtful inputs and especially for the opportunity to let us strengthen this manuscript. Hopefully, with the appropriate revisions, it can be accepted by the Journal.

Yours sincerely,

Hong-Yuan Huang, M.D.

Reviewer 2 Report

Thank you for the opportunity to review this manuscript which addresses the the study was designed to determine whether IL-18 expresses in the ectopic and/or eutopic endometrium, and to compare IL-18 system expression among the eutopic endometrium, ectopic endometrium, and corresponding myometrium in patients with adenomyosis. The study is very significant. Data of paired tissue samples with their homologous eutopic endometrium, ectopic endometrium, and corresponding normal myometrium were collected from 10 premenopausal women undergoing hysterectomy. Please find below some comments to assist you in strengthening your manuscript:

ABSTRACT

Add the aim of the study and the methods before the findings.

GENERAL

Line 47. Rephrase the sentence

Line 307. Rephrase the sentence

Most of the references are very old, it will be nice if current ones can be used.

Author Response

Point-by-point response to Reviewer 2

Thank you for the valuable comments. All of the changes we made in the manuscript were marked in red.

Comments:

ABSTRACT

  1. Add the aim of the study and the methods before the findings. Ans: Thank you. We added the study’s aim and methods in the revised manuscript.

GENERAL

  1. Line Rephrase the sentence

Ans: Thank you. We rephrased the sentence in the revised manuscript.

  1. Line Rephrase the sentence

Ans: Thank you. We rephrased the sentence in the revised manuscript.

  1. Most of the references are very old, it will be nice if current ones can be

Ans: Thank you. So far, the role of IL-18 in adenomyosis is unclear, and the associated papers were scanty. The references to general concepts were updated in the revised manuscript.

Thank you again for all the thoughtful inputs and especially for the opportunity to let us strengthen this manuscript. Hopefully, with the appropriate revisions, it can be accepted by the Journal.

Yours sincerely,

Hong-Yuan Huang, M.D.

Round 2

Reviewer 1 Report

4. In Figure 2, please illustrate with arrows or other means the areas and positions where antibody reactions are observed.

Ans: Thank you. The IL-18-positive reaction was illustrated with arrows in the revised manuscript.

Reviewer: Reviewer can see arrows in the areas that are not stained brown (Figure 2A D G H). Why do the authors indicate areas that are lightly stained when some areas are well-stained? Please correct the position of the arrow, replace it with a clear figure of the brown staining, or otherwise modify it.

5. In Figure 2, are the same magnifications used in the panels? A scale bar is needed.

Ans: Thank you. A scale bar was added in the revised manuscript.

Reviewer: The length of the scale bar is unknown. 100 µm? Authors should write in the figure legend. 

6. What morphological features are seen in endometrium and adenomyosis? Also, in which areas is IL-18 typically expressed? Please clarify.

Ans: Thank you. Ectopic endometrium is defined as the presence of islands of endometrial glands and stroma within the myometrium. A representational morphology of eutopic and ectopic endometrium is shown in Figure 2(A) and 2(B), respectively. IL-18 system expression was significantly localized in the cytoplasm of glandular and stromal cells. The statement was added to the revised manuscript.

Reviewer: At this scale, the internal structure of the cells would be difficult to understand. How did the authors demonstrate that the cytoplasm is predominantly stained with antibodies? If the authors claim that they are present in the cytoplasm, please provide a new figure.

9. All immunofluorescence data are of poor quality. Some panels are not in focus.

Ans: Thank you. We have revised the pictures.

Reviewer: The quality of the figures in the revised version is also poor, especially in the top-left and left-center figures. In the top-left and left-center figures, it is not clear where the DAPI-stained nucleus is located. Are the nuclei of the cells in this tissue observed a dispersed-staining pattern? Therefore, the DAPI staining may be incorrect or cells may be damaged. Please prepare correctly stained figures. Reviewer thinks the nuclear staining in the lower left figure is successful.

11. In Discussion, the experimental results obtained in this study should be discussed. Please add.

Ans: Thank you. The results presented in this study was addressed in Line 259-264, and further discussed in Line 314-322.

Reviewer: In line 334, please specify which Figures show the IL-18 dysregulation.

Round 3

Reviewer 1 Report

Comments:

1. There is a lack of information on what the areas indicated by the arrows (page 9, upper center panel) are and where they are distributed. In Figure 3 page 12, IL-18BP and IL-18R are expressed throughout the adenomyosis tissues, but are the results also reflected in Figure 2? The distribution of IL-18BP and IL-18R is also not indicated by arrows (Figure 2 page 9), and their expression and distribution are unknown. The authors should address to clarify where the IL-18 system is expressed and describe the detailed characteristics of that distribution. Arrow descriptions should be included in the legends.
